# Pastoral Conversion According to Franciszek Blachnicki in the Context of the Vatican's Instruction on Parishes in the Service of Evangelization (29 June 2020)

**Bogdan Jerzy Biela** 

Faculty of Theology, University of Silesia, 40-007 Katowice, Poland; bogdan.biela@us.edu.pl

**Abstract:** Endowed with an in-depth and insightful analysis of the Church's reality and the charism of reading the signs of the times, one of the greatest Polish pastoralists, founder of the Light-Life Movement, Fr. Franciszek Blachnicki (21 March 1921–27 February 1987) insistently postulated making the "Copernican turn" in the Church's saving ministry. On the hundredth anniversary of his birth, it is worth looking at the guiding principles of the Congregation for Clergy Pastoral conversion of the parish community in the service of the Church's evangelizing mission (29 June 2020) and analyze to what extent Blachnicki's concept of pastoral conversion regarding the renewal of the parish is still valid and estimate whether it can still be an inspiration in pastoral discussions on the realization of the Church *hic et nunc*.

**Keywords:** parish; evangelization; pastoral conversion; Franciszek Blachnicki



## 1. Introduction

The main burden of the Church's evangelizing responsibility rests primarily with the parish. Therefore, the statements of the Magisterium Ecclesiae, while pointing at the parish's importance in the realization of the Church, encourages at the same time "its more resolute renewal" (Paweł 1988). This postulate in the rapidly changing civilization processes is all the more urgent, because—as Pope Francis admits in the Apostolic Exhortation *Evangelii gaudium*—"the call to review and renew our parishes revises and renews our parishes has not yet borne sufficient fruit" (No. 28). Therefore, from the very beginning of his pontificate, Francis calls for a pastoral conversion that is the basis for the renewal of the pastoral mission of the Church (Franciszek 2013). This "pastoral conversion is one of the fundamental themes of the new phase of evangelization, which the Church is called today to promote, so that Christian communities may become more and more a driving force to encounter Christ" (ibid., point 287).

Pastoral conversion understood in this way was the goal of one of the greatest Polish pastoral theologians, the founder of the Light-Life Movement, now the Servant of God, Fr. Franciszek Blachnicki (1921–1987) (Blachnicki 2002). He was born in 1921 in Rybnik, in Upper Silesia. While taking part in the September Campaign on 20 September 1939, he was taken prisoner by the Germans. He made a successful escape attempt and began underground activities. In 1940, he was captured by the Gestapo once again and imprisoned in the German concentration camp in Auschwitz, where he stayed for 14 months. In 1942, he was sentenced to death by guillotine beheading for conspiratorial activity against the Third Reich. On June 17 in Katowice, waiting for the execution, he experienced his conversion. This event—as he recalled—was one of the most important events in his life. On 14 August 1942, after almost 5 months on death row, he was pardoned and the death penalty was changed to 10 years of hard prison. In the following years, until 1945, when he was freed by the American army, he stayed in Nazi camps and prisons in Racibórz, Rawicz, Börgermoor, Zwickau and Lengenfeld. After the end of World War II, he returned to Poland and entered the Silesian Theological Seminary in Krakow. On

25 June 1950, he was ordained a priest. In 1954 he organized the Oasis of God's Children retreat for the first time, using the closed retreats method. In 1957, he organized and led a social anti-alcohol initiative called the Temperance Crusade. Almost a thousand Catholic priests and over 100,000 lay people were involved in this initiative. The then authorities did not tolerate this type of activity, and on 29 August 1960 the initiative was liquidated. In March 1961, Fr. Franciszek was arrested for his activities and spent over 4 months in detention in Katowice. In October 1961, he started his studies at the Catholic University of Lublin. Based on the teachings of the Second Vatican Council, he developed a personalistic and ecclesiological concept of pastoral theology, as well as a concept and methodological foundations for fundamental catechetics and general pastoral theology. In 1973, after submitting his dissertation, Ecclesiological Deduction of Pastoral Theology, he was awarded a postdoctoral degree in pastoral theology. At that time, he was developing oasis retreats as an effective method of "new man pedagogy". He gradually applied the method of 15-day experience retreats to various groups of young people, adults and families. Over time, the oasis retreats developed into the evangelizing movement of the "Living Church", known from 1976 as the Light-Life Movement. In 1979, he initiated the Crusade for Human Liberation, the aim of which is to overcome alcoholism and other types of enslavement of modern man, and in 1980 Blachnicki launched "Ad Christum Redemptorem"—a plan of the great evangelization—the aim of which was to reach every person in Poland, sharing the Good News. Despite numerous bans and harassment from the communist authorities, the Light-Life Movement has become a sociological phenomenon throughout the so-called Eastern Bloc. It systematically developed (during martial law, almost 80,000 participants took part in "illegal oases" every year). The movement caught on in other countries as well.

During the "Solidarity" period (1980–1981), Fr. Blachnicki established the "Independent Christian Social Service" with the aim of promoting Christian social teaching and the liberation movement without violence, in line with the program "Truth-Cross-Liberation". He was in Rome when martial law was introduced in Poland. Unable to return to his homeland, in 1982 he settled in the Polish Marianum center in Carlsberg, Germany, and began organizing the International Light-Life Evangelization Center. Continuing his social and liberation work, he founded the Christian Service for the Liberation of Nations—an association gathering Poles and representatives of other nations of Central and Eastern Europe around the idea of internal sovereignty and unity of nations in the struggle for liberation. For this socio-liberating activity, he was harassed and repressed by the state authorities and slandered in the media in the People's Republic of Poland. He died suddenly in Carlsberg on 27 February 1987.

Blachnicki left a rich scientific, popular scientific, ascetic and formative heritage. His bibliography includes over 800 items. He is a spiritual father for the Light-Life Movement and for the communities of consecrated life connected with the movement: a female one—the Community of the Immaculate Mother of the Church, and a male one—the Community of Christ the Servant. John Paul II, after having learnt about the death of Fr. Blachnicki, in a telegram described him as a zealous apostle of the inner renewal of man and as the violence of the kingdom of God. On 1 April 2000, the posthumous remains of the Servant of God were brought from Carlsberg to the center of the Light-Life Movement in Krościenko n/Dunajec. On 9 December 1995, his beatification process began in Katowice. On 30 September 2015, Pope Francis signed a decree on the heroic virtues of the Servant of God (Wodarczyk 2008).

The founder of the Light-Life Movement was convinced that "the conciliar renewal will remain a fiction as long as it does not go down to the level of the parish, as long as we do not develop a model of the post-conciliar parish pastoral care" (Blachnicki 1995, p. 146). Endowed with the ability of an in-depth and insightful analysis of the Church's reality and the charism of reading the signs of the times, the author of the "Ecclesiological deduction of pastoral theology" (Blachnicki 1971) insisted on making a "Copernican turn" in the Church's saving ministry[1]. In a situation of accelerated "faith extinction" (not only

due to the COVID-19 pandemic)[2] it is worth reflecting on, on the hundredth anniversary of the theologian's birth, to what extent his teaching and postulates regarding parish renewal remain to date and whether they can still be an inspiration in pastoral discussions on the realization of the Church of *hic et nunc*. An important voice in this discussion is the instruction of 29 June 2020 of the Congregation for Clergy Pastoral conversion of the parish community in the service of the evangelizing mission of the Church, which takes up contemporary issues of shaping parish life (Kongregacja 2020).

## 2. Pastoral Conversion in the Instruction on Parishes

The traditional concept of the parish and parish ministry does not adequately meet the religious needs of contemporary people. This situation creates the need for the renewal of the parish and traditional forms of parish ministry (Benz 1967; Zulehner 1991; Slipek 2001; Polak 2007; Biela 2014). This was the main purpose of creating the Instruction of the Congregation for the Clergy: *the pastoral conversion of the parish community in the service of the evangelizing mission of the Church*[3]. In its introduction, it states, "a valuable opportunity for pastoral conversion that is essentially missionary. Parish communities will find herein a call to go out of themselves, offering instruments for reform, even structural, in a spirit of communion and collaboration, of encounter and closeness, of mercy and solicitude for the proclamation of the Gospel" (2). One of the main reasons this was written is certainly the situation of the parish in the West caused by the decreasing number of priests.

Regarding *the instruction for the pastoral conversion of the parish community in the service of the evangelizing mission of the Church*, the document's introduction quotes Pope Francis's words: "If something should rightly disturb us and trouble our consciences, it is the fact that so many of our brothers and sisters are living without the strength, light and consolation born of friendship with Jesus Christ, without a community of faith to support them, without meaning, and a goal in life" (3) (Franciszek 2013). Following the Pope's desires, the instruction encourages Christian communities to make a decisive missionary choice in the spirit of pastoral conversion (5). Therefore, in being creative, they may "seek new paths" (1), cultivate a "culture of encounter" (25), and become more of a driving force to encounter Christ (3). In order to achieve these aims, the parish needs a renewed dynamism to rediscover the vocation of every baptized person to be a disciple of Jesus and a missionary of the Gospel. Therefore, it is necessary to define perspectives that will allow for the renewal of "traditional" parish structures in the missionary key (20), because, especially today, the territory is no longer limited by geographical space, but by an environment in which everyone expresses their life through relationships and mutual service (16). Therefore, it is necessary for the parish to instruct people to read and meditate on the word of God through various ways of proclaiming the Gospel (22); use clear and understandable forms of communication that would speak of the Lord Jesus in accordance with the ever-changing testimony of the kerygma (21); and rediscover Christian initiation that gives birth to a new life. In view of these requirements, there is a need to designate mystagogical paths that relate to actual life (23). All this requires reflection not only on the new experience of the parish, but also of the ministry and mission of the priests who, together with the lay faithful, are required to be "the salt and light of the world" (cf. Mt 5: 13–14) and "a lamp on a lampstand" (cf. Mk 4: 21), which shows the face of an evangelizing community capable of properly reading the signs of the times and inviting us to bear consistent witness to the evangelical life (13).

All the elements of pastoral conversion presented above are contained in 26 points, in four chapters of the instruction: *Pastoral conversion; The parish in a contemporary context; The importance of the parish today; Mission, the main criterion for renewal*. The next (and fifth) chapter, "*Community of communities*", contains only seven points (27–33). The chapter emphasizes that the parish is first and foremost a specific community of the faithful, realized in communion and unity through the various members of the Body of Christ (27). He recalls the text from the Code of Canon Law, according to which the parish is a specific community of the faithful (can. 515; 518; 519). Francis's words from *Evangelii Gaudium*

indicate that the parish is a community of communities (28) (Franciszek 2013). Finally, in four points (30–33), the instruction indicates that the parish cannot be alien to the "spiritual and ecclesial style of Shrines"—true and proper "missionary outposts"—characterised as they are by their "spirit of welcome, their life of prayer and silence that renews the spirit, the celebration of the Sacrament of Reconciliation and their care for the poor" (30). The sixth chapter, "From the conversion of people to that of structures", indicates that the transformation of parish structures requires prior change of mentality and internal renewal, especially from those responsible for the pastoral care. It is they who, above all, should see the need for a reform of missionary pastoral care (35). This is why it would be appropriate to overcome both the "self-referential conception of the parish" and the "clericalization of the pastoral activity". This calls for the promotion of practices and models by which all the baptized, by virtue of the gift of the Holy Spirit and their infused chrisms, "become active participants of evangelisation, in the style and modality of an organic community, together with other Parish communities or at the diocesan level" (38).

The pastoral conversion of the parish in a missionary sense takes shape and is expressed in the gradual process of renewing structures (42). Therefore, the next six chapters of the manual—91 points in total—concern the transformation of structures, reorganization of parishes, the role of pastors and lay people in shaping parishes, etc. In fact, this part of the manual sparked the greatest discussion and criticism of the document, especially in Germany[4]. Most of the bishops, theologians and activists in the Catholic Church in Germany considered that the instruction was aimed at the "synodal path" and wanted to stop the initiative from achieving its aims[5]. For example, the vice-president of the German episcopate, Bishop Franz-Josef Bode, assessed that this document was "a strong brake on the motivation and appreciation of the ministry of lay people". Bishop of Essen Franz-Josef Overbeck pointed out that, despite the Vatican's instructions, he would still allow the laity to jointly manage parishes. By contrast, the bishop of Cologne, Cardinal Rainer Maria Woelkl, was positive about the instruction and highlighted its "proposals concerning the missionary awakening of the Church". In turn, cardinal Walter Kasper assessed that "German criticism completely ignores the correct intention of the instructions for conversion to the missionary ministry". Bishop Eichstätt Gregor Maria Hanke wrote that the Vatican's instruction focuses on the "concrete sacramentality" of the Church and not on structures. This document concerns sending the People of God to concrete evangelization in the parish[6].

Blachnicki, in the context of the conciliar ecclesiology of communion, which was officially confirmed by the Extraordinary Synod of Bishops in 1985, pointed to the role and importance of the local church, which is traditionally realized in the parish community. Certainly, due to various conditions, it is up to the particular churches to introduce some changes to the structure of the parish, without forgetting what the essence of the parish is. Therefore, in the context of the critical reception of the instruction on the parish in Germany, it is worth considering, in light of the guiding ideas of the instruction, how pastoral conversion is present in the pastoral ecclesiology of Fr. Blachnicki and meant to lead to the renewal of the parish—community of communities. This seems all the more justified because, as we have shown above, the issue of parish "community of communities" was treated in a brief and very general manner in the instruction. This is a fundamental problem, since the instruction Pope Francis states in point 29 is that "the call to review and renew our parishes has not yet sufficed to bring them nearer to people, to make them environments of living communion and participation, and to make them completely mission-oriented".

The pastoral conversion of the parish community, in terms of mission, takes shape and finds expression in a gradual process of a renewal of structures (42). Therefore, the next six chapters of the instruction—91 points in total—concern the transformation of structures, reorganization of the parish, the role of pastors and lay people in shaping parishes, etc. As a matter of fact, it was this part of the instruction which provoked the most intense discussions and criticism of the document, especially in Germany. Most bishops,

theologians and activists in the Catholic Church in Germany have come to the conclusion that the instruction is aimed at the "synodal path" and is meant to prevent its objectives, such as the transfer of ever-increasing administrative and pastoral power into the hands of the laity. Cardinal Walter Kasper, on the other hand, assessed that "German criticism completely ignores the actual intention of the instruction for conversion to the missionary ministry".

In this context, it is worth considering, in light of the guiding ideas of the instruction, how pastoral conversion is present in the pastoral ecclesiology of Fr. Blachnicki, which was devised to lead to the renewal of the parish, the "community of communities". This seems all the more justified because, as shown above, the issue of the parish understood as a "community of communities" has been treated briefly and very vaguely in the instruction. This issue is of pivotal importance, since the instruction, following Pope Francis, says in point 29 that "we must admit, though, that the call to review and renew our parishes has not yet sufficed to bring them nearer to people, to make them environments of living communion and participation, and to make them completely mission-oriented" (Franciszek 2013).

### 3. Pastoral Conversion in Blachnicki's Pastoral Ecclesiology

Studying the ecclesiology of Vatican II and the work of theologians of his time, Blachnicki showed that the concept of "koinonia", closely related to the idea of the People of God, the Mystical Body of Christ, the Church-sacrament and the family of God, appears as one of the definitions of the rich reality adopted in the conciliar ecclesiology covered by the name "Church". This "new image of the Church" turned out to be valuable, as it could become a "leading image" for pastoral theology and pastoral care, clearly defining the goal, the subject, means and methods of the Church's *cura pastoralis*[7].

Summarizing Blachnicki's analyses, it can be stated that koinonia is, in essence, the participation of people in the life of the Triune God caused by God's self-giving. It is therefore koinonia with the Father through the Son in the Holy Spirit. The resulting community of life with God, which can be described as a vertical koinonia, consequently leads to a fraternal interpersonal community, which can be called a horizontal koinonia. Both these aspects of the community are inseparable; however, the former has priority over the latter, because the union with God through Christ and in the Holy Spirit creates the objective foundation of interpersonal unity. The trinitarian genesis of koinonia justifies its personal character. Both vertically and horizontally, community is created by actualization of personal attitudes and interpersonal relationships, and ultimately by actualization of the attitude summarized in the fact that a person possesses himself in giving himself to the other person, which is essential for the personal being. In this way, the person, fully realizing himself, enters at the same time into fellowship with other persons. The essence of koinonia can therefore be defined by the concept of agape, meaning personal love consisting of the gift of self. At the same time, this love is the essence of a person's life, which is expressed in community.

The koinonia, which is the essence of the Church, should also be defined more precisely by the concept of the sacrament. The Church is a community-sacrament in the time that has come after the original community, lost by sin, and before the final community, which will be fully realized only in the eschatological order of the resurrection. The Church as the sacramental community (community-sacrament) must be a visible sign of the invisible reality that constitutes her essence. This essence is the self-gift of God to people through Christ, accepted in the Holy Spirit, which constitutes the unity of these people with Christ and with each other. The sign of the self-gift of God to people through Christ is the whole order of proclamation of the word and of the administration of the sacraments in the Church. One should add to this—also as a sign—the ministry of a hierarchical office that guarantees the authenticity of the relationship between the word and the sacrament with the historical and glorified Christ. The sign of people accepting God's self-gift in the Holy Spirit is the whole order of faith and love. The Church understood in this way as a vertical and horizontal community is manifested and realized above all in the sign of the liturgical

assembly, especially of the Eucharistic one. In this sign, all the elements that make up the community of the Church are present, and if the external signs correspond to the internal ones, personal attitudes of the participants, which can be summarized as faith and love, this assembly actualizes the Church as a community. The Eucharistic assembly must therefore be given a central position in the work of realization of the Church.

The personal and sacramental character of the Church community shows that it must be realized in specific local communities, because a community is made up of a set of real, realized interpersonal relationships, contrary to relating a group of people to an abstract or ideal community in the sense of, for example, the universal Church. The Church existing in the local community is at the same time a sign of the universal Church, in which this Church is truly present, whereas the universal Church is a community of local churches. It follows that the realization of the Church consists of the creation and development of concrete and local communities that, for their part, are once again focused and expressed in the sign of the Eucharistic communion. The local community is therefore the proper "area" for the realization of the Church. Since the Church is a sign and instrument of uniting people with God and with one another, she is this sacrament only insofar as she expresses herself in the sign of a concrete and realized community. Such a community is therefore an instrument for granting the graces of salvation to those who are still outside of it. In other words, the sacramental community, and therefore the Church realized in a specific local community, is also a means of salvific mediation. The Church, if she is a living Church, is a life-giving Church as well. This means that concern for the most intense realization of the Church in communities, and thus concern for the realization of the Church as a community-sacrament, also ensures its extensive growth most effectively (Blachnicki 1971)[8].

As the result, Blachnicki formulated the principle of the Church's operation[9]:

"The salvific mediation of the Church (i.e., pastoral ministry) must be exercised for this purpose and in such a way as to make God's self-surrender present in Christ in word and sacrament, and to condition the free acceptance of this surrender in mutual self-giving in the Holy Spirit through faith and charity, for the realization of a community in the vertical aspect (with God) and the horizontal aspect (with the brethren), in the visible and effective sign of the Eucharistic assembly and the local community in communion with the universal Church". (ibid., pp. 454–55)

Apart from far-reaching conclusions for the theory of the life and activity of the Church, this principle—according to Blachnicki—gives, above all, the possibility of realizing the vision of a "new pastoral ministry", which the pre-conciliar ecclesiology could not create. The pre-conciliar ecclesiology was built on the sociological and legal vision of the Church and, by virtue of that fact, it could have been a source of inspiration for certain administrative and disciplinary actions. Moreover, the ecclesiology in question presented the pastoral ministry unilaterally in line with the scheme "shepherd—flock", which was connected with clericalization and the passive attitude of the faithful who slowly turned into lapsed Catholics in fact outside the Church (Blachnicki 1972).

The result of these reflections was a proposal to move from pastoral ministry (*cura animarum*) to evangelization. Blachnicki described it as the "Copernican phrase":

"We have not yet made the fundamental transition from pastoral care to evangelization. Meanwhile, it is the Copernican turn that we must make in our consciousness [ . . . ]. At the moment, as a matter of fact, the Conciliar concept—especially the one that found its expression in the Exhortation *Evangelii nuntiandi*—and the concept that is still enduring by gravity are conflicting ones. We are not even aware of this conflict. The situation, in general, is that we are trying to cover all that was and still is with a "conciliar cap", but we do not change anything. A new "lid" covers the old content. This is how the current situation looks, even in the field of terminology itself. The term "evangelization" has become a legitimate term and one keeps talking about it all the time [ . . . ]. However, when we slide

under the surface of these slogans, we will see exactly the same reality that was formerly called pastoral care[10].

In view of the above, Blachnicki began to lean towards the "catechumenal" concept of pastoral care, postulating that the institution of catechumenate should become a source of the fundamental rules of life and development of the Christian community (cf. Congregatio pro Cultu Divino 1972). He justifies his point of view as follows: "Since the goal of pastoral work is to build a local community, and the goal of the catechumenate is to educate and lead to life in this community, then one could legitimately speak of this very concept of pastoral work" (Blachnicki 1976):

"If parishes are introduced to evangelization understood as the method of—let us keep using this term—pastoral care, then each parish must understand that its normal function is to carry out evangelization. Consequently, a catechumenate will arise in the parish as a fruit of evangelization. A new system of formation for mature faith must now be established in every parish, which will fill in the shortcomings of catechesis. Catechesis is one of the elements of the catechumenate. Catechesis will, of course, remain in place, as it is needed and useful, and no one will question it. Catechesis must have its basis and conditions which it does not possess in itself, and only then will it be effective. The catechumenate educates people who are capable of witness to another active evangelization. Then, the parish will change from a static to a dynamic one and will not only stop the shrinkage process but also reverse it. Various evangelizing impulses will come out of it and the parish will be able to be the Living Church"[11].

The founder of the oasis movement did not focus only on theoretical considerations, but at the same time undertook a search for practical applications of the "catechumenal concept of pastoral care". The fruit of these reflections was a personal commitment to creating organizational structures that enliven the local community. A special tool for the formation of individuals towards Christian maturity, as well as an effective project of transforming the parish into a community of communities, was the system of formation created by Blachnicki in the Light-Life Movement. The stages of the educational process, i.e., the path of the new man, in the pedagogy of Blachnicki's movement were defined as catechumenal or deuterocatechumenal ones, because the founder consciously referred to the catechumenate system developed in the first centuries of the Church, within which the process of initiation into a full and mature Christian life was carried out[12]. "The pedagogy of the new man of the Light-Life Movement is therefore in its present form nothing else than a fully developed system of catechumenal formation, adapted to the requirements of our times and adjusted to be embedded in the structure of ordinary parish life"[13].

According to the catechumenate pattern, the path to Christian maturity in the Light-Life Movement is divided into three stages: evangelization, the proper catechumenate and the period of the so-called mystagogy. The stage of evangelization in the deuterocatechumenal system of the Light-Life Movement precedes the first degree oasis retreat. The purpose of this stage is to lead to a religious awakening through a personal encounter with Christ and to accept Him by faith in the Holy Spirit as the Lord and Savior. This is done by an evangelization retreat or an individual evangelization[14]. Evangelization is followed by a post-evangelization period aimed at verifying the motives for conversion and strengthening the decision to accept Christ. The oasis of New Life of the first degree is the moment of transition to the proper deuterocatechumenate, i.e., joining a group of students who want to enter the path leading to mature faith. The first year of deuterocatechumenal formation, as a continuation of the first degree oasis, focuses all formation around the word of God and faith. For this year, a program of the so-called ten steps to Christian maturity is scheduled[15]. The second degree oasis, followed by the second year of deuterocatechumenal formation, is a period of in-depth liturgical and sacramental initiation. This period ends with the renewal of the baptismal covenant during the Easter Triduum celebrated as a retreat. The third degree oasis, on the other hand, corresponds to the mystagogy period

in the catechumenate program. Its theme "Ecclesia Mater—Mater Ecclesiae" begins the time of a deeper introduction into the mystery and life of the Church community. The third year of the deuterocatechumenate, following the oasis of the third degree, also focuses on the themes of charism, vocation and diaconia. In this way, entering the community of the Church is prepared by taking up a specific diaconia in it, in accordance with the charism possessed, which is to be the lasting fruit of all deuterocatechumenal formation and, at the same time, the completion of basic formation in the pedagogy of the new person (Blachnicki 1987e).

The entire system of primary formation in the Light-Life Movement is based on a small community that becomes a primary formation group. This principle applies both to year-round formation in parish groups and to the Oasis of New Life holiday retreats (Blachnicki 1980). Retreats conducted in accordance with the oasis method and formation work during the year in parish oasis communities are two basic, cyclically repeated components of the formation system of the Light-Life Movement. At each stage of formation, apart from biblical study, catechesis and individual work, community elements are realized, such as Eucharist, devotion to the Word of God and group meetings. Blachnicki emphasized that the specific goal of deuterocatechumenal formation is to emphasize and update the interpersonal relationships that constitute the essence of Christian life. Therefore, the entire formation system of the Light-Life Movement is not limited only to presenting the theory of Christian life, but above all is to create conditions for experiencing this life. The fifteen-day holiday retreat plays a special role. The central idea behind the oasis retreat experience is to discover the source of a truly Christian life, which is Christ. This is done not only through intellectual knowledge, but above all through a personal decision to personally accept Jesus Christ as one's Lord and Savior. The place of this meeting is not only the word, prayer and the sacraments, but also the relationship of love with another human being. Therefore, "the basic commandment of the oasis is AGAPE—selfless personal love, realized in a double relationship—to Christ and others, members of the oasis community" (Blachnicki 1987f, p. 43). The structure of the oasis retreats is intended as a kind of school of life permeated with selfless love, service to one's neighbor and transgressing one's own selfishness. The "agape" climate should become a value of each participant internalized strongly enough to be able to survive when confronted with the everyday realities of the living environment. "This basic experience of an oasis, that constitutes its essence, becomes a conscious—to varying degrees—lifestyle of its participants. The experience continues to grow, becomes permanent and deepens in small communities of "permanent oases" in parishes and other environments. In this way, these communities can become a leaven and a tool for building up a living Church" (ibid., p. 64).

The Light-Life Movement and the formation system offered by it is built on the charism of meeting Christ and other people. This encounter becomes a source of apostolic dynamism and creates a need for commitment and service (ibid., pp. 69–71). The crowning achievement of the deuterocatechumenal period of formation in the Light-Life Movement is therefore taking responsibility for the movement and the community of the local Church. This is done through the decision to get involved in various diaconies of the movement, which are an important manifestation of community life which confirms personal commitment. Therefore, Blachnicki was convinced that the pastoral vision brought by and implemented by the Light-Life Movement is a fundamental vision in relation to the traditional model of pastoral ministry. He was also aware that the "Copernican turn" in pastoral ministry is some kind of revolution, extremely difficult, requiring long-term and formative activities, especially among priests (Blachnicki 1991). This new vision, derived from the ecclesiology of communion—which our pastor ist draws particular attention to—inspires, above all, the life of the local communities of the Church. Therefore, the Light-Life Movement created by him was to be, above all, a means and a tool to achieve the basic goal of a living local community[16].

### 4. Pastoral Conversion in the Realization of the "Community of Communities"

The ideas expressed in the instruction on parishes can be summarized in a single sentence: the way to convert parish structures is the conversion of people. This is exactly what Blachnicki had in mind when he formulated his *cura pastoralis*:

"In pastoral theology, one speaks of the so-called activist concept of pastoral care. There are various actions, but there is no comprehensive vision, there is no understanding that pastoral care is building a community [ . . . ] Pastors must see that the goal of their workis to educate a new man in the community of the People of God and build this community, then all actions are a means to an end" (Blachnicki 2002).

In view of the above, one of the key and basic conditions for the implementation of the communion concept of pastoral care was for Blachnicki pastor's awareness of an integral and organic vision of building a local community:

"The pastor is the man who is the builder able to embrace the whole picture. He must have a holistic vision. He cannot succumb to activism and care only for how to reconcile various projects, to fit it all in time so that everything runs smoothly. He needs to know the direction of his actions, at what stage a particular community that is to build a living Church is, and what processes must be initiated. Coordinating and guiding this process according to a certain vision is the main task of the presbyter" (Blachnicki 1991).

Additionally, this is a necessary condition—let us add—if a pastoral conversion of the parish community is to take place. According to Blachnicki, the traditional approach to pastoral care does not embrace a "holistic concept" of pastoral care (Blachnicki 1991). There is either an extremely activist approach, focusing on a number of different "actions", which are most often based on the fear of "vacuum", or a problem-based approach that tries to comprehensively capture a problem and strive to solve it. As a consequence, priests either experience deficiency and insufficiency of the activities undertaken or, on the other hand, suffer from physical and mental exhaustion (Blachnicki 1979b). Therefore, a holistic approach is needed, the starting point of which is the goal or ideal you want to achieve. The pastor embraces the entire development process that is to lead to the realization of the ideal of the Church. From this perspective, he assesses the meaning and purposefulness of individual actions and tries to solve problems. In light of the conciliar ecclesiology of communion, the main goal of pastoral efforts should be the realization of the Church community. The pastoral effort should therefore be directed to the renewal of the parish in such a way that the parish becomes a community (see Blachnicki 1979a).

However, a parish can only be transformed into a community by creating small groups within it and through small groups. The parish must therefore be a "community of communities". This means that there is no direct way to transform parish members into a community. "Community is the result of relations between individuals—and these must be concrete relations, not some vague idea of unity" (Blachnicki 1987d). Therefore, information and organization alone are not enough (Blachnicki n.d.). Above all, there is a need for participation and a testimony that can spark life. Therefore, in the first place there must be a leaven in the parish—a living cell—that is, a Christian community that can satisfy the needs of its members in the field of Christian life. However, for a group to be a living cell of the Church, that is, a sign and a form of its realization, it must meet certain conditions. First of all, it must be built on the foundation which is Christ. The ecclesial community is the result of a personal encounter with Christ, which constitutes the essence of communion in the vertical dimension. One should persist in this vertical community if one wishes to create an interpersonal community.

The community must therefore be a place of meeting with God. The first element that brings the group together is the word of God. It must be the standard of every act. When this word of God gathers and unites the group, it is a sign of the Church, because the Church is a community of those who listen to the word of God and keep it. Another

element is prayer. The Church is a community of prayers who, having the Holy Spirit, respond to God's call. The response to the word of God, which is visible in the life of the communities, is also metanoia, as a constant striving to live according to the Gospel. Ecclesial communities must represent an evangelical lifestyle expressed in an ordered hierarchy of values. This lifestyle is also a basic form of testimony to the environment. If within a group there is a joint effort to change life, then it is a sign of God's people as a community of those who live their lives in the Holy Spirit. The life of such a group must also be linked to the liturgy, especially the Eucharistic liturgy, which is the full sign of the Church and at the same time an element that creates and forms the group.

Ecclesial communities must offer their members the environment and place to experience agape—evangelical love—expressed in mutual bearing and forgiveness as well as in service to each other and to the community. One of the essential features of the ecclesial community is therefore the diaconia. It begins in a small group where everyone contributes to building it, and then moves to the wider community, undertaking some service to a parish, diocese, or community. Such a group reveals the Church that is a community of services and charisms. It is closely connected with participation in the missionary consciousness of the Church, expressed in the sense of responsibility for the salvation of the world and in undertaking various evangelization initiatives. Therefore, a small group must be open to other communities in order to enter into the great unity of the Church of Christ. Therefore, the union through the priest with the bishop makes the group to be built into the local Church and through the bishop into union with the Holy See and the universal Church. This striving for unity is a special sign of the charismatic inspiration of a movement or community rooted in the Holy Spirit (Blachnicki n.d.) See (Blachnicki 1987c, pp. 22–24).

If such communities were to be able to take over various tasks in the local community, creating a living Church—a community of communities—a sine qua non condition would be to change the traditional model of a parish. According to Blachnicki, the parish model, derived from the reform of the Council of Trent, is today unsustainable, because the Tridentine principium of parish creation is the division of the parish into an active and passive principium, and into a teaching and listening Church. There is a two-way relationship to be observed: "me–them". There is no awareness of "we", and only when such awareness appears can one say that the Church is a community, both vertically and horizontally. Vertically, because both the pastor and the faithful are jointly responsible for building the Church, and horizontally, in multiple contacts between the faithful, which are actualized in specific communities.

In the post-Tridentine model of the parish, which includes a certain group of people, there is a constantly growing number of people who do not practice their faith or declare openly that they are non-believers. There is also a second category of parishioners—those who do practice their faith. However, these practitioners do not have much of an influence on non-practitioners because there is no ecclesial "we" in their consciousness. Generally speaking, there is a tendency to quantitatively increase the number of people who do not practice and to diminish the influence of the pastor. The entire parish model is therefore a static one, there is no missionary dynamic in it. There is a permanent location where people come to satisfy their religious needs, and the sphere of influence of priests is actually confined to the circle of those who come to Church, catechetical room or parish office (Blachnicki n.d.). Therefore, there is a need for a dynamic missionary model characterized by the awareness of "we". This, in turn, is possible only in communities that create interpersonal relationships. Thus, small groups must be this intermediate stage in building a community bond in this larger team, which is the parish. Additionally, then, thanks to these communities, and thanks to their apostolic witness and commitment, it is possible to lead the whole parish to renew itself in its communal character. It is worth emphasizing, following Blachnicki, that in this process of building a parish community an indispensable element is the Eucharistic assembly, which is, in a way, a "living model" in which the community can recognize itself as the Church. Therefore, everything must be done so

that the Eucharist is lived in its fullest dimension as an effective sign of the Church of the community (ibid., pp. 103–4).

In a community parish we can enumerate seven circles. The first circle—to be found on the periphery—includes both the baptized and the non-baptized. It is a missionary circle that is a task for others, embracing it with their prayer, concern and apostolic dynamism. The second circle consists of faithful practitioners who are in the Church on the basis of a "me–them" relationship, treating the parish as a place that provides religious services. This circle must be an area of formation and education too so that these faithful may be witnesses of Christ for others. The next circle is made up of small groups. It includes people from the previous two circles who make up the catechumenal groups. It is a place of systematic formation, which replenishes the shortcomings of traditional religious formation and introduces life in the ecclesial community. In these groups, the process of maturing the ecclesial consciousness of "we" takes place. These communities are already living cells that influence the entire parish. From these groups, another two circles of diaconal communities are formed. The first of them are responsible for specific environments, e.g., a workplace, a block of flats or a residential quarter. Other diaconal groups assume responsibility for certain vital functions of the Church that must be performed in each local community—for example, a diaconia for liturgical matters, the media and for charitable matters. Due to a large number of groups in the community parish, it is necessary to establish a ministry team (diaconia team)—an apostolic council that interacts and coordinates activities of the various groups. The last circle—located in the very heart of the parish—is the permanent deacon, which includes the parish priest, vicar and a group of lay people completely devoted to the local community. This group should be a sign for the whole parish of what the parish is to be educated to and how it is to develop. Therefore, the testimony of this group is extremely important. The outlined model of the parish meets, according to Blachnicki, the requirements of the Second Vatican Council. It is also a missionary model whose dynamics and inspiration do not stop at any circle. One can see a movement from the center to the periphery—there are the dynamics of evangelization, and from the periphery to the center—which is the dynamics of growth. In this way, a "new parish" is created: a community of communities (ibid., pp. 104–5).

> "The dilemma that still arises occasionally whether the Church should concentrate on creating true ecclesial communities—thus accepting the situation of the diaspora—or whether the Church's concern should go towards embracing the widest possible masses, in this context it loses its raison d'être. The conciliar perspective is unequivocal. This is primarily due to the essence of the Church, which by its very nature is missionary (DM 1: 2), and therefore for everyone—for the world. The Church, however, will be an effective sacrament of salvation for all the more if she will be herself, trying to reveal her essence in the local community. For this reason, the principle of the primacy of intensive and qualitative realization of the Church over a "quantitative" one should be adopted. The more intensely the Church realizes her essence by becoming a fraternal community, the more extensively it will influence other people as the sacrament of salvation" (Blachnicki 1971).

Based on the conciliar vision of the Church, according to Blachnicki, a comprehensive and organic concept of *cura pastoralis* is possible, understood as the process of building the Church in the local community. The process of realizing the Church begins with evangelization, which initiates the processes of liberation in people and in a specific environment. The next stage is introduction—through catechumenal formation—to life in the ecclesial community. Therefore, for Blachnicki, the main conclusion arising from the fact that the Church is realized in specific local communities is the postulate that the entire pastoral ministry should be so reconstructed that it serves the formation of basic communities of Christian life within the parish[17]. It is therefore necessary to introduce the model of a community parish into the implementation stage. In order to solve the dilemma arising in this context—namely, how to implement the model of the post-conciliar

parish—if it is impossible to build a living Church community without the existence of mature ecclesial communities, the founder of the Light-Life Movement pointed to the role and opportunity offered by various movements based on a holistic, catechumenal formation (See Sędek 2002).

The role and significance of the *cura pastoralis* developed by Blachnicki, based on the model of evangelizing and catechumenical pastoral ministry, which is to lead to the implementation of the missionary parish of the community of communities, is confirmed by the post-conciliar teaching of the Church[18] and the emerging contemporary projects of parish renewal and parish pastoral ministry[19]. Therefore, it can be concluded that the founder of the Light-Life Movement in his scientific creativity and charismatic intuitions was ahead of his time, becoming a kind of prophetic lumen in the assimilation and implementation of Vatican II and the post-conciliar teaching of the Magisterium Ecclaesie. The main conclusion resulting from the fact that the Church is realized in specific, local communities was for Blachnicki the postulate that the entire pastoral ministry should be so reconstructed that it serves the formation of basic communities of Christian life. Their role in shaping parish life—especially in the post-pandemic time—cannot be overstated. The concept of pastoral ecclesiology offered by Blachnicki is therefore still valid, and the resulting vision of *cura pastoralis*—especially in the practice of parish life—still calls for understanding, exploration and consistent implementation, all the more so as it fully fits both in theory and practice in the pastoral conversion of the contemporary Church postulated by Pope Francis in the Exhortation *Evangelii gaudium* (cf. points 25–33), and is it expressed in the latest instruction issued by the Congregation for the Clergy concerning the renewal of the parish community.

**Funding:** The article is subsidised by the following: MEiN subsidy (the Ministry of Education and Science)—research potential (provisions of the Head of the Theological Sciences Institute).

**Institutional Review Board Statement:** Not applicable.

**Informed Consent Statement:** Not applicable.

**Data Availability Statement:** The study does not report any data. For details concerning quoted works, see the bibliography.

**Conflicts of Interest:** The author declares no conflict of interest.

## Notes

[1] On the meaning of theological thought and the works undertaken, see (Biela 2008, pp. 297–310; Buchta 2009) Ks. Franciszek Blachnicki—katechetyk i pastoralista. W dwudziestą rocznicę śmierci Sługi Bożego.

[2] Of course, the reason for "extinguishing faith" is not just the COVID-19 pandemic. Certainly, the internal problems of the Church, such as the loss of trust due to the clerical abuse crisis, contributed to the decline of religious faith as well. It is a fact, however, that due to the pandemic, the Church faced new problems and unique pastoral challenges (impossibility of gathering in communities, closed churches, lack of formation and sacramental life, openness to new forms of transmission of the faith and the related pastoral conversion). See: (Sawa 2020).

[3] A.Ripa, Undersecretary of the Congregation for the Clergy, presenting the document on 21 July 2020, emphasized that the purpose of the instructions is to propose some ways to help in the dynamics of "outreach", which, according to the Pope's wishes, should characterize all parishes. Therefore, there is a need for a pastoral conversion, which must extend to all the baptized since each of them must take part in the evangelizing mission of the Church. Therefore, to respond to the significant social and cultural changes taking place in the world, it is necessary to move from a conservative, closed pastoral care to an outgoing, missionary pastoral work. See: (Watykańska Instrukcja o parafiach na służbie ewangelizacji n.d.).

[4] See: (W Niemczech trwa debata nt. Instrukcji o parafiach n.d.; Kościół w Niemczech o watykańskiej instrukcji o parafii n.d.).

[5] The Synodal Way began in December 2019. The process continues with almost equal participation of bishops and laity. Its aim is to bring about changes in four areas: sexual morality (especially in the field of homosexuality); the role of women (especially in the diaconate and priesthood); celibacy; and the division of power between clergy and laity. The President of the Pontifical Council for Promoting Christian Unity, Cardinal Kurt Koch was negative about the entire project.According to the cardinal, today's renewal requires a first-line focus on new evangelization. Therefore, it is not so much about reforming structures, but about the authentic proclamation of the Gospel, as Pope Francis wrote in his famous "Letter to the Pilgrim People in Germany" of June 2019. Additionally, Cardinal Koch warned against the constant search for "novelty", for the real novelty in the Catholic

Church is the Savior Himself—and it is He who must always be at the center of all efforts for renewal. See: (Kard. Koch upomina Niemców: „Odnowa to stawianie w centrum Chrystusa" n.d.).

6   According to the former president of the Pontifical Council for Promoting Christian Unity, Germany is setting wrong priorities: the constant discussion of celibacy, the priesthood of women and the responsibility of the laity lead to fewer priestly vocations. According to Cardinal Kasper, the priest's overall responsibility for the parish is theologically justified. Bishop Eichstätt Gregor Maria Hanke also wrote that the Vatican's instruction focuses on the "concrete sacramentality" of the Church and not on structures. This document is about sending the People of God to concrete evangelization in the parish. See: (Kościół w Niemczech o watykańskiej instrukcji o parafii n.d.).

7   See: (Blachnicki 1992). The book is a part of his post-doctoral dissertation *Eklezjologiczna dedukcja teologii pastoralnej*, in: (Blachnicki 1971). See (Biela 1993, pp. 20–55).

8   See: (Ratzinger 2003, 2009, 2012, 2016; Kasper 1986, 1988; Napierała 1985).

9   Based on the "New Image of the Church", Blachnicki formulated the principle of the life of the Church, which defines how the Church is to be realized in accordance with the will of Christ and its nature in order to ensure internal and external growth: The principle of the Church's life is "koinonia—that is, realized in a visible sign of the ministry of the word and sacrament and the social unity of faith and love—the community of people with Christ and with himself in the Holy Spirit, who, as one and the same person in Christ and all members of the Church constitutes the invisible essence of this community". See: (Blachnicki 1971, p. 444). This formula, called by Blachnicki the formal principle of pastoral theology, is of great importance for pastoral care, as it allows for a scientific and theological assessment whether the activity of the Church at a given historical moment and in a given environment is in line with the essence of the Church and its mission. On the other hand, it allows us to positively define the action of the Church as to its purpose, essential elements, and attributes. See: (Blachnicki 1972, pp. 436–37).

10  (Blachnicki 1991, pp. 11–12). According to Blachnicki, "The term pastoral theology is still a bit misleading. I believe that it should be replaced with the name theology of evangelization, because pastoral theology covers only one fraction of evangelization (when sheep are in the sheepfold, they need to be taken care of). The concept of evangelization is much broader because it includes all the fields to be gained. It conveys all the dynamics and the expansion. Then you have to immediately set goals, tasks, means and methods from the theoretical point of view. Within this broad vision, there will then be a place for Seelsorge; taking care of those souls who have already believed and need systematic care". Ibidem, pp. 14–15.

11  (Blachnicki 1991, p. 14). According to Blachnicki, this process of transformation begins rather slowly in the Church, despite the fact that the formulation of the Council that the entire Church is missionary is essential and crucial: "So wherever the Church is to be found, there must be a missionary dynamic, there must be a missionary attitude, a missionary tendency. The problem of missions is not primarily a problem of the mission countries where missionaries are sent, but it is the problem of our parishes, which must be missionary, because if they are to be the Church, there must be missionary dynamics in them". Ibidem, p. 13.

12  The receptive subjects of this formation are usually baptized people who have already lived through some—albeit rudimentary—catechumenate, especially in connection with the so-called preparation for the First Holy Communion and as part of school catechesis; there fore, the deuterocatechumenate, being repeated catechumenate, has been adopted. See: (Blachnicki 1987e, p. 37).

13  The resumption of the order of the catechumenate in the post-conciliar document Ordo initiationis christianae adultorum (1972) was adopted by Blachnicki as a kind of catalyst accelerating the process of shaping the pedagogy of the new man in the Light-Life Movement. See: ibid.

14  Evangelization retreats as part of the Light-Life Movement take place according to the plan "Ad Christum Redemptorem", inspired by the encyclical of John Paul II Redemptor hominis and the experience of the American evangelization movement "Campus Crusade for Christ". See: (Blachnicki 1999).

15  The original program of "Ten steps towards Christian maturity" developed by Blachnicki (Jesus Christ, Mary Immaculate, the Holy Spirit, Church, the Word of God, prayer, liturgy, testimony, new culture, agape) includes small group meetings conducted by the method of "evangelical conversation", the participant's own work, based on a notebook specially prepared for this purpose, and devotions of the Word of God, gathering members of several formation groups.

16  (Blachnicki 1987a). On the basis of Paul VI's exhortation *Evangelii nuntiandi* and the encyclical of John Paul II *Redemptor hominis*, Blachnicki developed a large-scale plan for the evangelization of parish communities. See: (Blachnicki 1988).

17  (Blachnicki 1976). "This truth is very comforting for us, because if, for example, in a parish of a few thousand or several dozen thousand people, several small groups of several dozen people arise and these people meet, sharing the word of God, pray, give testimony, it really creates a new living environment [ . . . ]. Today, small cells scattered throughout the Church give the Church the power to renew the world because, especially in these small groups, the Church becomes a sacrament—not only a sign, but also an instrument of uniting people with God and with itself". See: (Blachnicki 1987d).

18  Cf. (Paweł 1988); Potrzeba i zadania nowej ewangelizacji na przełomie II i III Tysiąclecia Chrześcijaństwa, in: (Wydawnictwo Pallottinum 2001, p. 43). For example, the pastoral vision of the parish ministry presented in the resolutions of the Second Synod of the Archdiocese of Katowice is based on the model of pastoral ministry developed by Blachnicki. See: Duszpasterstwo w Kościele katowickim, in: (Wsłuchani 2016, pp. 2–8).

19  See: (Żądło 1994, 1999; Alberich and Binz 1995, pp. 173–87). Some of these projects are very popular and quite widely used in pastoral practice—e.g., the project "New Parish Image" of the Movement for a Better World (Cappellaro 2000) or "Ewangelizacyjne

komórki parafialne"(Macchioni 1997)." Small group communities" implemented in Protestant churches are noteworthy too. Their strategy is that parish building is not so much about having small groups within its structure, but about having the parish community composed of small groups. This strategy has been used successfully in the Saddleback Baptist Church in California. It is now the fastest growing evangelical community in American history. Within 25 years, the number of believers increased from 0 to 25,000. See (Warren 2005, p. 9). In 2016, the number of small groups for adults on which the Church in Saddlebackwas founded was over six and a half thousand. See (Gladen 2016, p. 16).

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
