# Peer review of "Pastoral Conversion According to Franciszek Blachnicki in the Context of the Vatican’s Instruction on Parishes in the Service of Evangelization (29 June 2020)"

_religions, doi:10.3390/rel12090762_

Round 1

Reviewer 1 Report

This is an interesting article which introduces a figure who is not well-known in the English-speaking world, Franciszek Blachnick. It would be good to include a short paragraph on his life, including the fact that he was a prisoner at Auschwitz and later defended human rights in communist Poland. 

# 39: reference to 'faith extinction' - is this the author's own phrase? Clarify further and specify briefly what particular challenges for parishes have emerged during the pandemic. Also, when making a strong statement like '"faith extinction" (not only due to the COVID-19 pandemic) ...' it should be acknowledged that internal church problems, e.g., the loss of trust due to the clerical abuse crisis, have also contributed to decline of religious faith.  

Are there any similarities between the communities Blachnick proposes and the base ecclesial communities of liberation theology in their focus on the bible being read in small communities?

How do these small communities within parishes manage to cross boundaries of class and ensure diversity in the Christian community? (See reference to a block of flats or a residential quarter in line 318 as a locus for a community)

Further, how do these communities within communities accommodate diverse theological opinions and diverse life experiences, including those for whom aspects of church teaching can be experienced as alienating?

The paragraph beginning #91, dealing with the reception of the Congregation's Instruction in Germany could be a little more developed to reveal the author's own position and to explicate more clearly the significance of Blachnick's contribution in the light of the critical reception.  

The Abstract needs some editing: 'analyze to what extent Blachnick's concept of pastoral conversion regarding the renewal of the parish is still  valid and estimate whether it can still be an inspiration ...'

There are some minor corrections needed, including the following:               

  • Latin titles and book titles should be in italics
  • # 30 - single quotation marks within a quotation
  • # 49 - remove semi-colon
  • # 92 - concern
  • Footnotes need editing for consistency of formatting
  • # 55 & 56 need editing
  • # 137: just use the surname without the term 'servant of God'. It would be sufficient just to use this term in the short biography of Blachnick

Author Response

To begin with, thank you for reading the text carefully and for the constructive and specific comments, which I will take into account while editing and correcting the article.

This is an interesting article which introduces a figure who is not well-known in the English-speaking world, Franciszek Blachnick. It would be good to include a short paragraph on his life, including the fact that he was a prisoner at Auschwitz and later defended human rights in communist Poland. 

- You are absolutely right, especially for a foreign-language reader, the article needs a bio of Blachnicki, which I will include in the article.

# 39: reference to 'faith extinction' - is this the author's own phrase? Clarify further and specify briefly what particular challenges for parishes have emerged during the pandemic. Also, when making a strong statement like '"faith extinction" (not only due to the COVID-19 pandemic) ...' it should be acknowledged that internal church problems, e.g., the loss of trust due to the clerical abuse crisis, have also contributed to decline of religious faith.  

- The phrase "faith extinction" is my expression, which - as you suggest - should be supplemented with causes other than the coronavirus.

Are there any similarities between the communities Blachnick proposes and the base ecclesial communities of liberation theology in their focus on the bible being read in small communities?

- There is a close similarity between the so-called base communities that grew in Latin America and the communities proposed by Blachnicki. However, Blachnicki refers to the base communities that are strictly ecclesial in nature, and discussed by, inter alia, Pope Paul VI in "Evangelii Nuntiandia" or John Paul II in "Christifideles laici". When it comes to the theology of liberation developed in Latin America, of which Blachnicki (being repressed by the communist authorities) was a radical opponent, which is why he created the Polish version of the “non violence” theology of liberation (based on internal freedom). It was supposed to be disseminated by "Liberation of Man Crusade" and the “Christian Service for the Liberation of Nations”, two ministries created by Blachnicki.

How do these small communities within parishes manage to cross boundaries of class and ensure diversity in the Christian community? (See reference to a block of flats or a residential quarter in line 318 as a locus for a community)

- Basic communities gathering in houses - according to Blachnicki's assumption - are available to everyone, because the main goal of a community parish is precisely to reach the periphery of the parish by groups of committed Christians. In such a group, everyone can find their place, as well as ultimately religious formation and specific service for the parish community. Such communities are "Domestic Church" groups - one of the branches of the Light-Life Movement. Blachnicki established them in order to evangelize a specific environment, which is families.

Further, how do these communities within communities accommodate diverse theological opinions and diverse life experiences, including those for whom aspects of church teaching can be experienced as alienating?

- It is difficult to give a precise answer to this question, as it depends on the specific group and its leaders. Generally, for example, in the Domestic Church, apart from praying and teaching, there is time for sharing problems and doubts.

The paragraph beginning #91, dealing with the reception of the Congregation's Instruction in Germany could be a little more developed to reveal the author's own position and to explicate more clearly the significance of Blachnick's contribution in the light of the critical reception.  

- I fully agree with the suggestion to explain the importance of Blachnicki's contribution in the context of the critical reception of the Instruction on the parish in Germany.  

There are some minor corrections needed, including the following:               

  • Latin titles and book titles should be in italics
  • # 30 - single quotation marks within a quotation
  • # 49 - remove semi-colon
  • # 92 - concern
  • Footnotes need editing for consistency of formatting
  • # 55 & 56 need editing
  • # 137: just use the surname without the term 'servant of God'. It would be sufficient just to use this term in the short biography of Blachnicki

Thank you for indicating the minor errors, which will of course be corrected.

Reviewer 2 Report

Dear author!

First of all, thank you for letting me read your manuscript and learn something about Franciszek Blachnicki and his pastoral work. You have chosen a very special question, to discuss a pastoral concept with a Vatican Instruction, and to hermeneutise it into the lines of a post-conciliar ecclesiology.

After reading, I would like to say a few words about the methodology and then little about selected examples. 

Your methodology is very pillar-based. You briefly introduce the Instruction, then in step two you discuss post-conciliar ecclesiology and in step three pastoral ministry. A methodical interconnection takes place between shadows two and three at certain points; basically, however, the three parts stand very much side by side.
Then they have to shorten strongly due to the specification of the textual version, which, however, leads to methodological distortions. Example: I understand on p.5 their approach to briefly rot the pre-conciliar ecclesiology, but their abridgement leads to their textual statements being wrong in content. Pre-conciliar ecclesiology knew a Body of Christ theology, which is not mentioned, and which the Council - with the support of Pope Pius XII - substantially prepared. "Mediator Dei" was a seminal encyclical. This theology had a great part in seeing ecclesiology not only in legal terms, and this even before the Council. Think of Roman Guardini, his theology and the statement that the Church awakens in souls. This was a guiding principle for the movements before the Council.
When presenting the parish and the concept according to Franciszek Blachnicki, I wondered in the meantime whether they were presenting a classic parish or a movimento. Should it be a parish, you are receiving the theology of the 70s; all the great processes after that are avoided in the idealistic presentation and are thus missing. Should it be a movimento, I would look for the theology of communio.
By the way: can a parish really mitigate the consequences of secularisation with pastoral effort? You have formulated this thesis and in the Western European / US-American discourse it cannot be communicated in this way.

I come to a few selected examples, although I could name many:
On p.5 you conceive a "on the one hand" - "on the other hand" pair of terms without this really helping in terms of content. You present the local church one-sidedly as the image of the authentic church. You would have to change this linguistically. Moreover, I miss a reference to the great theological debate (Ratzinger-Kasper) in 1993 in the high weight they attach to the universal local church discourse.
In academic writing, one does not use the form 1st person singular or plural. I noticed that more often.
You write of the post-postridentine parish. What is that supposed to be? The Council said a lot about the people of God, but nothing substantively about the structure or nature of the parish. Even then, it was not possible to write a uniform document for the universal Church. 

Finally, dear author, I would advise you to withdraw your manuscript and revise it again. You should discuss the critically requested theological remarks, perhaps in an upper seminar or in your studies. If this text were to appear under their name in this way, they would not really recommend themselves in international discourse and their concern to make Franciszek Blachnicki and his pastoral work known internationally would hardly be served.

However, please feel free to comment on my remarks. In any case, I wish you and your academic work good luck and maximum success!  

Author Response

Thank you for your critical reading of my manuscript and for your response.

  1. After reading, I would like to say a few words about the methodology and then little about selected examples. 

Your methodology is very pillar-based. You briefly introduce the Instruction, then in step two you discuss post-conciliar ecclesiology and in step three pastoral ministry. A methodical interconnection takes place between shadows two and three at certain points; basically, however, the three parts stand very much side by side.
Then they have to shorten strongly due to the specification of the textual version, which, however, leads to methodological distortions. Example: I understand on p.5 their approach to briefly rot the pre-conciliar ecclesiology, but their abridgement leads to their textual statements being wrong in content. Pre-conciliar ecclesiology knew a Body of Christ theology, which is not mentioned, and which the Council - with the support of Pope Pius XII - substantially prepared. "Mediator Dei" was a seminal encyclical. This theology had a great part in seeing ecclesiology not only in legal terms, and this even before the Council. Think of Roman Guardini, his theology and the statement that the Church awakens in souls. This was a guiding principle for the movements before the Council.
When presenting the parish and the concept according to Franciszek Blachnicki, I wondered in the meantime whether they were presenting a classic parish or a movimento. Should it be a parish, you are receiving the theology of the 70s; all the great processes after that are avoided in the idealistic presentation and are thus missing. Should it be a movimento, I would look for the theology of communio.
By the way: can a parish really mitigate the consequences of secularisation with pastoral effort? You have formulated this thesis and in the Western European / US-American discourse it cannot be communicated in this way.

In trying to respond to the comments presented - as you have correctly noticed - due to the size of the article and the issues enumerated in the title, it is difficult to present an interesting theological discourse on pre-conciliar and conciliar ecclesiology, since this topic goes beyond the scope of my text. Of course, the ecclesiology of Vatican II was largely based on the idea of the Body of Christ and the People of God, developed before the Council. This is what Blachnicki strongly referred to in his research, which is why I made this brief comment: "Studying the ecclesiology of Vatican II and the work of theologians of his time, Blachnicki showed that the concept of "koinonia", closely related to the idea of the People of God, the Mystical Body of Christ, the Church-sacrament and the family of God, appears as one of the definitions of the rich reality adopted in the conciliar ecclesiology covered by the name "Church". This "new image of the Church" turned out to be valuable, as it could become a "leading image" for pastoral theology and pastoral care, clearly defining the goal, the subject, means, and methods of the Church's cura pastoralis”. Obviously, the purpose of the text is not “to briefly rot the pre-conciliar ecclesiology”, but to show that in the present situation of the Church, the accentuation is on the ecclesiology of communio, which was officially emphasized by the Extraordinary Synod of Bishops in 1985, which is also referred to in the Vatican instruction on parishes.

In view of the above, when speaking of the parish, Blachnicki emphasized its community dimension in accordance with the definition of the Code of Canon Law of 1983. (Can. 515), as opposed to the definition of a parish as part of a territory (cf. Code of Canon Law of 1917, can. 216). The parish understood in this way - as Blachnicki showed - leads to important consequences for its realization as a community. So it is not some idealistic vision/presentation of the parish, but a dynamic community of communities. An exemplary implementation of  the vision is the Light-Life Movement founded by Blachnicki and a number of other movements that undertake the renewal of a parish according to the ecclesiology of communio (see footnotes 52 and 53). To give you yet another example from the US, Saddleback Church in California is developing dynamically based on the aforementioned vision  (footnote 53). This example shows that the parish and the community of communities to a large extent is capable of "mitigating the consequences of secularisation with pastoral effort".

  1. I come to a few selected examples, although I could name many:
    On p.5 you conceive a "on the one hand" - "on the other hand" pair of terms without this really helping in terms of content. You present the local church one-sidedly as the image of the authentic church. You would have to change this linguistically. Moreover, I miss a reference to the great theological debate (Ratzinger-Kasper) in 1993 in the high weight they attach to the universal local church discourse.
    In academic writing, one does not use the form 1st person singular or plural. I noticed that more often.

As you have rightly observed the rhetorical figure: "on the one hand" - "on the other hand" had not contributed to the clarity of my manuscript, which is why it was edited accordingly. Of course, it would be interesting to present a fascinating theological debate about the local church (incl. Ratzinger-Kasper), but in my opinion it is fundamental theology that is at stake in this respect and as such the suggested discussion  would consequently obscure the narrative of the article. I believe one might elaborate upon the issue in a separate article on Blachnicki’s concept of ecclesiology, which apparently goes along the teaching of Ratzinger and, to a large extent, Kasper. Likewise, it is correct to note that in academic texts the use of the 1st person, both singular or plural, is unacceptable. The mistake had somehow been made and appropriate corrections were itroduced.

  1. You write of the post-postridentine parish. What is that supposed to be? The Council said a lot about the people of God, but nothing substantively about the structure or nature of the parish. Even then, it was not possible to write a uniform document for the universal Church. 

As for the use of the expression "post-Tridentine parish", offered by Blachnicki,  this model of parish,  derived from the reform of the Council of Trent, is still being implemented within the Church. The main point is that the Tridentine principium of creating parishes, that is, the division of the parish into an active and passive principium, is untenable. In this model, there is a two-way relationship: "me-them". There is no awareness of "we", and only when such awareness appears can one say that the Church is a community, both vertically and horizontally.  

You are absolutely right that the documents of Vatican II did not devote any document to the parish. We have only 33 texts about the parish, 10 of which are mentioned on the margin of the doctrine of the universal Church (eg KK 26, KL 42, DB 23, 30-32, DK 6, 8-9, DA 10). It must be said, however, that although the Council speaks of a parish in an incidental way, it can nevertheless define the position of Vatican II on many issues which constitute the basis for the proper direction of parish renewal and parish pastoral ministry. Reflection on the parish depends on the way the Church is understood. Consequently, the structure and character of the parish were discussed, among others, by plenary synods on evangelization or the role of the laity in the Church (Exhortations: "Evangelii nuntiandia", "Christifideles laici"). John Paul II left a huge legacy of teaching about the parish as well. All these statements, as I have indicated, confirm the vision of shaping the parish according to Blachnicki.

  1. Finally, dear author, I would advise you to withdraw your manuscript and revise it again. You should discuss the critically requested theological remarks, perhaps in an upper seminar or in your studies. If this text were to appear under their name in this way, they would not really recommend themselves in international discourse and their concern to make Franciszek Blachnicki and his pastoral work known internationally would hardly be served.

As for the last remark, I have to admit that I had already started a discussion with several of my fellow professors (especially pastoralists) from several theological faculties, including foreign ones, and they all said that the article was worth publishing not only because of the topicality of the issues raised, but also due to the unique role that Blachnicki played (not only in Poland) in the implementation of - as he himself called - the "Living Church".

Thank you so much once again for your stimulating comments and suggestions.